# Application of Sebum Lipidomics to Biomarkers Discovery in Neurodegenerative Diseases

**DOI:** 10.3390/metabo11120819

**Published:** 2021-11-29

**Authors:** Stefania Briganti, Mauro Truglio, Antonella Angiolillo, Salvatore Lombardo, Deborah Leccese, Emanuela Camera, Mauro Picardo, Alfonso Di Costanzo

**Affiliations:** 1Laboratory of Cutaneous Physiopathology, San Gallicano Dermatological Institute—IRCCS, Via Elio Chianesi 53, 00144 Rome, Italy; stefania.briganti@ifo.gov.it (S.B.); mauro.truglio@ifo.gov.it (M.T.); mauro.picardo@ifo.gov.it (M.P.); 2Centre for Research and Training in Medicine of Aging, Department of Medicine and Health Science “V. Tiberio”, University of Molise, Via De Santis, 86100 Campobasso, Italy; angiolillo@unimol.it (A.A.); slombardo92@virgilio.it (S.L.); d.leccese@studenti.unimol.it (D.L.); alfonso.dicostanzo@unimol.it (A.D.C.)

**Keywords:** fatty acidomics, branched chain fatty acids, cholesterol, squalene, triglycerides, wax esters, sebaceous gland, Parkinson’s disease, Alzheimer’s disease, cognitive impairment

## Abstract

Lipidomics is strategic in the discovery of biomarkers of neurodegenerative diseases (NDDs). The skin surface lipidome bears the potential to provide biomarker candidates in the detection of pathological processes occurring in distal organs. We investigated the sebum composition to search diagnostic and, possibly, prognostic, biomarkers of Alzheimer’s disease (AD) and Parkinson’s disease (PD). The observational study included 64 subjects: 20 characterized as “probable AD with documented decline”, 20 as “clinically established PD”, and 24 healthy subjects (HS) of comparable age. The analysis of sebum by GCMS and TLC retrieved the amounts (µg) of 41 free fatty acids (FFAs), 7 fatty alcohols (FOHs), vitamin E, cholesterol, squalene, and total triglycerides (TGs) and wax esters (WEs). Distributions of sebum lipids in NDDs and healthy conditions were investigated with multivariate ANOVA-simultaneous component analysis (ASCA). The deranged sebum composition associated with the PD group showed incretion of most composing lipids compared to HS, whereas only two lipid species (vitamin E and FOH14:0) were discriminant of AD samples and presented lower levels than HS sebum. Thus, sebum lipid biosynthetic pathways are differently affected in PD and AD. The characteristic sebum bio-signatures detected support the value of sebum lipidomics in the biomarkers search in NDDs.

## 1. Introduction

Neurodegenerative diseases (NDDs) such as Alzheimer’s disease (AD) and Parkinson’s disease (PD) are accounted among the main causes of chronic disability and mortality. The prevalence of the two NDDs is predicted to increase in the future [1,2,3]. Both pathologies are extremely elusive in their onset and development. It is now clear that when the neurodegeneration becomes clinically manifest, brain damage has reached an advanced stage. In preclinical AD, individuals have measurable biomarkers of brain changes, but they have not developed symptoms yet [4]; in PD, classic motor symptoms appear only when more than a half of neurons in the *substantia nigra* (SN) undergo overt degeneration [5]. Thus, it is crucial to make an early diagnosis to adopt appropriate strategies in the prevention of neuronal death.

Etiopathogenetic mechanisms causing neurodegeneration are very heterogeneous and complex. Unfortunately, many of them are still unknown. This represents a major constraint in the individuation of sensitive and specific diagnostic biomarkers, or in the development of effective therapies. The available biomarkers are only ancillary to the clinical diagnosis and many of them are obtained through invasive and risky and/or costly and time-consuming methods such as lumbar puncture, cerebral FDG-PET and amyloid imaging used for AD [6]. Thus, there is a need to identify new sensitive and specific biomarkers, obtainable with fast and moderately-to-minimally invasive methods on accessible samples such as blood, saliva, urine and sebum. In the last few years, thanks to technologies such as mass spectrometry (MS), lipidomics is gaining attention in the processes of biomarkers discovery. [7]. This science can assess lipid profiles in cells, tissues and biological fluids, and assists the identification of altered metabolic lipid pathways in many pathologies, including NDDs [8,9]. For example, recent lipidomics studies underlined a significant increase of some lipids in the blood of PD patients compared to controls [10,11,12,13]. Lipidomic analysis in blood has also been carried out in AD patients. Several lipids that seem to play an essential role in the AD pathogenesis have been proposed as biomarkers [14,15,16].

PD and the skin are related in a number of ways, which include abnormalities of the disease itself that are detectable clinically. An increased prevalence of skin diseases, such as seborrhoeic dermatitis (SD) and melanoma, has been reported among PD patients [17,18]. SD occurs in 50–60% of PD patients and is therefore accounted as a premotor feature of the disease [19,20].

Considering the key role of sebum composition in SD, together with the evidence that dermatological conditions may precede the onset of typical motor symptoms by years, the lipidomic analysis of sebum may allow the identification of biomarkers that precede the motor features of PD.

Recent studies have proposed the use of sebum to screen for PD. Interestingly, investigations of volatile metabolites arising from skin areas rich in sebaceous glands (SGs) revealed the existence of a differential volatile profile between control subjects and PD patients [21,22]. Moreover, a difference in the composition of sebum in PD compared to control subjects has been reported in a study based on untargeted LC-MS analysis [23]. In addition, AD is not exclusively neurological, but rather involves multiple tissues and organs. The abnormalities in metabolic and biochemical processes described in affected brains may be reflected in the skin and lay the foundation for specific dermatological manifestations [24]. Interestingly, seborrhoeic keratosis, which is an age-related skin disease prevalent in regions of the body with high density of SGs, is associated with the over-expression of amyloid precursor protein (APP), a key player in the pathogenesis of senile AD [25].

Despite the number of the lipidomic studies performed in NDD patients, it has never been investigated whether elemental components of sebum may be altered compared to healthy controls. Sebum represents an easily accessible biological fluid, thus examinable through non-invasive methods that could provide new NDD biomarkers and assist the identification of new therapeutic targets.

This study aimed to evaluate the differences in the profiles of elemental components of sebum in two NDD conditions, i.e., PD and AD, to discover a lipid pattern and/or metabolic lipid pathways that can assist the non-invasive and timely diagnosis of NDDs.

Here, we applied gas chromatography-mass spectrometry (GCMS) profiling of sebum building blocks, mainly free fatty acids (FFAs) and fatty alcohols (FOHs), integrated with squalene, cholesterol, vitamin E, triglycerides, and wax esters, the latter ones determined by thin layer chromatography (TLC), to investigate differences between unaffected controls and subjects with AD or PD. In addition, findings of the study can contribute to the molecular pathomechanisms underlying disturbed sebogenesis in PD.

## 2. Results

Sebum samples were collected from a total of 24 HS (10 F, 14 M), 20 AD (11 F, 9 M), and 20 PD (4 F, 16 M) subjects. While HS and AD presented rather balanced F/M ratio, in the PD group the number of M was 4-fold that of F. Average, SD and statistical significance of the evaluated parameters are reported in Table 1. Average age of the three groups was in the range between 70–80 years and was rather comparable among groups. BMI, smoking habits, and alcohol intake were similar in the three groups. Nevertheless, M were leaning toward normal weight in the PD group. In agreement with the literature, AD group was characterized by a significantly lower education status compared to HS or PD. In general education status was higher in males; the difference reached the significance in the AD group. SER values were significantly higher in the PD group compared to HS. Consistently unequal SER values in the two genders was observed in all groups. The differences between F and M reached statistical significance in the HS group only, wherein M had an average SER almost two-fold higher than F. Nevertheless, the differences observed in the PD group were to be considered under the influence of the prevalent gender among these patients.

### 2.1. Amounts of Lipid Species in the Sebum of HS, AD and PD Subjects

Absolute amounts of lipids were quantified in the sebum absorbed in 30 min from two tapes, which account for a skin surface of 11.2 cm^2^. Table 2 reports the average absolute amounts (µg) of individual lipid species (FFAs, FOHs, squalene, cholesterol, and vitamin E) or lipid classes (TGs, and WEs) in the three groups. Between HS and AD samples no relevant differences were found. PD group was characterized by a general significant increase (*p* ≤ 0.05) of lipid components compared to HS, suggesting the hyperactivation of sebum secretion. Moreover, the evaluated indexes such as desaturation indexes, namely Fatty Acid Desaturase 2 (FADS2), expressed by the ratio C17:1/C17:0, and stearoyl-CoA desaturase-1 (SCD1), assessed by the ratio C18:1/C18:0, as well as the C24:0/C18:0 ratio, illustrative of the elongation of FAs, were modified in PD, as shown in Appendix A. In particular, the index of the SCD1 desaturation pathway and the exogenous to endogenous FFAs balance were, respectively, increased and decreased at a significant extent in PD. Due to the MUFAs-TGs and MUFA-WEs interdependency in sebaceous lipid pathways, we evaluated the MUFAs to TGs ratio and MUFAs to WEs ratios in the HS and NDD conditions. The MUFAs/TGs ratio was 1.8- and 3.2-fold higher in AD and PD, respectively, compared to HS; differences were close to significance (*p* = 0.017) and significant (*p* = 0.002) in AD and PD, respectively (data not shown). The MUFAs/WEs ratio was 1.5- and 2.0-fold higher in AD and PD, respectively, compared to HS; divergence from HS were significant in both AD and PD, being *p* = 0.008, and *p*= 0.002, respectively (data not shown).

### 2.2. Correlation among Age, BMI, SER, and Sebum Lipid Components

Correlation analysis is useful in understanding the relationship among lipid metabolites in the biological changes involved in NDDs in association with factors implicated in sebogenic activity. Spearman’s rank correlation was used to analyze the correlation insisting among age, BMI, SER, individual sebum lipids, and indexes of FA metabolism, i.e., FADS2 (C16:1/C16:0 and C17:1/C17:0 ratios), SCD1 (C18:1/C18:0 ratio), linoleate/sapienate ratio (C18:2/C16:1), and elongation (C24:0/C18:0). Based on shared metabolic pathways, FFAs have been grouped according to their main characteristics as reported in the Appendix A. Significance of correlations was defined according to the restrictive *p*-value cut-off of 0.017 (Bonferroni correction). Figure 1 depicts the correlation matrices for HS, AD, and PD, in the panels (a), (b), and (c), respectively, whereas correlations and *p*-values are reported in Appendix A. The correlation matrix generated in HS group showed that age correlated inversely with the levels of vitamin E delivered onto the skin. While BMI showed to be unrelated to other parameters, SER was positively and significantly correlated with all the evaluated sebum indexes, except eSCFAs, FOHs, vitamin E, WEs, and the C17:1/C17:0 and C24:0/C18:0 ratios. The C18:2/C16:1 ratio, proposed as an indicator of exogenous source of FFAs opposed to that of endogenous origin, was negatively correlated with SER. In AD group SER was positively correlated with several sebum lipid components, in particular eBCFAs, oBCFAs, oSCFAs, MUFAs, squalene, and C17:1/C17:0 ratio, suggesting the preservation of a balanced composition of sebum. The negative correlation between SER and C18:2/C16:1 ratio was accentuated in AD compared to HS. Amounts of TGs presented several positive and significant correlations with abundance of summed sebaceous-type FFAs in HS. Such correlations were absent in both AD and PD, suggesting that the TGs composition could be uncoupled with the abundance of FFAs in these NDDs. Interestingly, while the abundance of TGs correlated with that of squalene and WEs in PD subjects, consistently with HS, TGs lacked association with any other sebum-parameter in AD. Interestingly, in PD subjects we found a generalized lack of correlation between SER and sebum components, indicating that the increased sebum excretion in PD associates with a disturbed synthesis and/or turnover of sebaceous lipids. Moreover, SER showed a statistically significant inverse correlation with age in the PD group only. Age was also negatively correlated with the amounts of TGs, and WEs. Noteworthy, BMI was inversely correlated with eBCFAs, oBCFAs, oSCFAs, and the C24:0/C18:0 ratio. Thus, lower BMI in PD appeared to lean towards accentuation of sebaceous character of lipid profiles.

### 2.3. Chemometric Discrimination of Sebum in AD and PD from That of HS

ASCA analysis was performed on both absolute amounts (µg) and relative values (weight/weight percentage, *w*/*w*%) obtained after normalization of the absolute lipid amounts with respect to the total sebum quantity measured gravimetrically. Employing ASCA to describe the analytical data and to find different lipid profiles, we were able to minimize the effect of sebum amount in the observed differences of lipid species in HS, AD and PD groups. ANOVA partitioning of the variance for the foreheads after data matrix normalization evidenced that the effect of NDD condition accounted for 5.02% of the total variability, that of gender for 8.70%, the interaction for 1.18%, and the residuals for the remaining 85.10% (data not shown). Permutation tests allowed to assess that both the effect of NDD (*p* < 0.0001) and that of gender (*p* = 0.0001) were significant for this data set (plots not shown). Therefore, both factors were studied by ASCA. In the overall data classified according to the NDD factor, i.e., HS, AD, and PD, there was a significant effect due to the NDD conditions (Appendix A). The scores along the two principal components (PCs) SCA1, and SCA2, which represented 91.5% and 8.5% of the total variability, respectively, were represented in the score plot (a) in the Appendix A. To interpret the score trends, the loadings were investigated according to their length and direction as depicted in the loadings plot (b) in the Appendix A. Both SCA1 and SCA2 scores had positive sign for the majority of PD samples. In contrast, the majority of AD samples were characterized by negative values of both SCA1 and SCA2. HS samples fell in the quadrant with negative and positive SCA1 and SCA2, respectively. The scores along the SCA1 discriminated mainly the PD group from HS and AD, whereas the scores along the SCA2 discriminated mainly AD from HS and PD. The loadings plot reports the loadings contributing significantly to discrimination. The loadings were color coded as described in the legend. The loadings with a full line and a dotted line were referred to the first and the second SCA, respectively. A wide spectrum of sebum lipids presented higher amounts in PD subjects, in particular MUFAs, terminally branched FFAs, long chain FOHs, TGs, WEs, and squalene. Fewer species were discriminant of AD samples; in detail vitamin E and FOH14:0 showed lower and higher levels, respectively, in AD. These results need to be interpreted taking into account the weight of gender, as it was not equally distributed in the three groups. This is particularly relevant in the context of human sebum since it, in turn, is differently abundant in the two genders. As shown in the score plot (**a**) represented in the longitudinal view in the Appendix A, F and M were clearly discriminated based on the loading scores. The majority of F and M subjects, which are labeled on the x axis by their ID, showed negative and positive scores, respectively, independently on the healthy or NDD conditions. From the loadings plot in the panel (**b**) of the Appendix A it is apparent that most of the lipid species had significantly higher values in M. Two lipid species, i.e., FOHC16:0 and FOHC18:0, were significantly higher in F. To attempt minimization of the drifting effects of the higher SER in M, the same multivariate analysis was performed on amounts relative to sebum weight (*w*/*w*%). The outcome showed that *w*/*w*% profiles were gender-driven similarly to absolute amounts (data not shown). Altogether, these observations suggested to perform the comparisons of AD or PD versus controls, separately in F and M subjects. As shown in the Appendix A, permutation tests demonstrated that the effect of the NDD condition was significant in both F (*p* = 0.0002) and M (*p* = 0.0001) absolute data sets. Therefore, the NDD factor was successively studied by ASCA in the F and M sex, separately. ASCA performed in F and M are reported in Figure 2 and Figure 3, respectively.

As shown in Figure 2 and Appendix A, the comparison of data in F subjects revealed that the scores along the principal component SCA1 discriminated AD from HS, whereas the four F in the PD group were discriminated along the principal component SCA2. One control subject (HS-020) had a SC1 score closer to the ones of the AD group. A considerable number of variables was different in F AD, in particular the majority of MUFAs, oBCFAs (11Me-C13:0, 12Me-C15:0, 13Me-C15:0), eSCFA (C12:0, C14:0, C24:0), oSCFAs (C13:0, C21:0, C23:0, C25:0), and cholesterol. In contrast, the amounts of a few lipids, i.e., vitamin E, C14:1, and C16:0 in F, had higher and lower abundance in the PD and the AD groups, respectively. Both groups were characterized by lower levels of FOH18:0 compared to HS.

As shown in Figure 3 and Appendix A, the comparison of data in M subjects revealed that the scores along the principal component SCA1 discriminated PD from AD and HS. The loadings plot indicated that most sebum components were significantly higher in PD patients. Two PD patients fell in the quadrant of the majority of HS. In contrast, five controls (HS-038, HS-040, HS-043, HS-051, and HS-052) localized in the quadrant of positive SCA1 and SCA2 together with PD. The scores along the principal component SCA2 discriminated M in the AD group from M in HS and PD groups. The loadings plot showed that only a few lipids were significantly changed in AD; in particular, FOHC14:0 and FOHC16:0 were significantly higher and 9Me-C12:0 together with TGs were significantly lower in AD.

## 3. Discussion

Our study has undertaken quantitative profiling of sebum in PD and AD patients and unaffected controls (HS). The sebometric analysis showed that sebum weight and thus SER of PD patients were significantly higher than HS, while AD sebum parameters were similar to those of HS. Consistent with the literature, PD was associated with hyper-seborrhea [26], which may account for the high incidence of seborrheic dermatitis in this NDD [27]. In adult population, sebum production declines with ageing in both males and females. It has been estimated that every 10 years after adolescence sebum decreases by 23% and 32% in males and females, respectively. This implies that the older the population, the wider the difference in the SER values between males and females. We found appreciable sebum production in all groups, in contrast with the alleged negligible amounts detectable in men and postmenopausal women in their 60s and 70s, respectively [28]. In the present study, we found differences in SER between males and females compatible with the different rates of decrement with aging, even if significant differences in SER between males and females were found only in the HS group. The sebum quantity was correlated with levels of sebum-specific lipid species, such as squalene and WEs. The significant difference found in the PD group is probably due to the male prevalence among the patients. The age-dependent decline in the sebum output, with the number of SGs remaining approximately unchanged, has been associated with the cutaneous xerosis of the elderly [29,30,31]. Seborrhea in PD seems to be an exception in the age-related modifications of the SG activity in several respects. Elevation of SER in PD can be due to delayed decline of the SG activity or to hyperactivation of the SG due to PD pathomechanisms sustaining sebogenesis, including derangement of regulatory factors in the neuroendocrine system in the CNS, and dysbiosis. In addition, pharmacological therapies and hormonal status may influence SER [32,33]. Clarifying the temporal changes of excessive sebum production might have important implications in the detection of PD ahead of deterioration of the motor functions.

In contrast, values of SER in AD appeared to be aligned to those observed in the HS group, in both genders. To the best of our knowledge, this is the first study addressing sebum in AD. Moreover, despite the higher frequency and lifetime risk of AD in women than in men, an important contributor to this sex difference is that women live longer than men. Thus, to maximize the development of current and future treatments and interventions across the AD spectrum, sex and gender differences in AD must be better understood and measured [34].

Females with AD displayed different levels in BCFAs in sebum, where they display relatively higher abundance compared to other biofluids [35]. Potentially, BCFAs arise from multiple pathways, i.e., de novo synthesis from branched amino acids, metabolism of skin microbes, and diet [36,37]. Branched amino acids, i.e., leucine, isoleucine and valine, have been shown to occur at lower level in AD sera [38]. Further investigations are needed to assess the crosstalk between circulating and skin amino acids.

Among the 51 elemental lipid compounds analyzed by GCMS and two sebum classes, i.e., TGs, and WEs, determined by TLC (Table 1), we found profiles of abundance that characterized both AD and PD sebum fingerprints. This is the first lipidomics analysis of sebum elemental components, i.e., FFAs, FAOHs, squalene, cholesterol, and vitamin E together with total TGs and WEs in patients with NDDs. Previous lipidomics studies have been performed on blood or brain, and significant alterations in some lipid concentrations have been discovered. For example, higher levels of the SCFAs C16:0 and C18:0 (C16:0 is the same found increased in PD sebum) have been observed in lipid rafts from the frontal cortex of PD patients compared to controls [39]. Mielke et al. [10] found that levels of ceramide, monohexosylceramides and lactosylceramides were higher in PD patients’ blood than in controls and these higher levels were associated with worse cognition. In addition, Chan et al. [11] underlined more elevated levels of the ganglioside GM3 in plasma of PD cases compared to controls. Blood lipidomics studies in AD have shown many and heterogeneous lipid alterations compared to HS, including changes in sphingolipids, fatty acids, phospholipids, and TGs levels [15]. For example, concentrations of long chain cholesteryl esters (CEs) were lower in AD patients compared to individuals with mild cognitive impairment (MCI) and controls [14].

Sebum is physiologically composed of FFAs, TGs, WEs, squalene, cholesterol, and cholesterol esters [40]. Sapienic acid (C16:1n-10) is an almost exclusive sebaceous-type FA and the most abundant one among the sebaceous MUFAs [41,42,43]: this feature is confirmed in PD patients’ sebum. It deploys appreciable antibacterial and antifungal properties [44,45,46]. It is important to note that, within the increased SCFAs in PD sebum, there is also palmitic acid (C16:0) from which derives sapienic acid, by FADS2-catalyzed desaturation [46]. In analogy to palmitate, C17:0 is a substrate of FADS2 [47]. Thus, both the C16:1/C16:0 and the C17:1/C17:0 ratios are indexes for the FADS2 pathway in the SG [48]. Like sapienic acid, oleic acid (18:1n-9) is also increased: this is one of the main sebaceous MUFA with the strongest antibacterial and antifungal activity [49]. TGs account for the highest *w*/*w*% in sebum. TGs are formed upon esterification of glycerol with FAs. Abundance and composition of TGs in sebum are associated to the variety of sebaceous-type FFAs. The association between sebaceous FFAs and TGs observed in unaffected controls was apparently deregulated in both NDDs. Together with the observation of higher MUFAs/TGs and MUFAs/WEs ratios in both NDDs, these features were convergent in AD and PD sebum. Another crucial compound increased in PD is squalene, well known as an intermediate in cholesterol biosynthesis. Characteristically, squalene is not converted to cholesterol in the SG; thus, it could be considered a marker for sebocytes differentiation and sebogenesis [50]. Quantitative and qualitative alterations of lipids found in PD sebum may have several explanations. More likely, the main regulatory mechanism is the neuroendocrine system in CNS. Indeed, SGs express several receptors for corticotropin-releasing hormone (CRH), α-melanocyte-stimulating hormone (α-MSH), β-endorphin, vasoactive intestinal polypeptide (VIP), neuropeptide Y and calcitonin gene related peptide (CGRP). They seem to be regulated by the hypothalamus–pituitary–skin (HPS) pathway (conceptually similar to the hypothalamic–pituitary–adrenal axis) [28,51]. A hypothesis is that alterations in the HPS axis and in the levels of neuropeptides occurring in PD, likely due to central dopamine depletion, lead to the modification of SER and sebum lipids composition. Specifically, the altered neuroendocrine system may up-regulate transcription factors of lipogenic enzymes such as peroxisome proliferator-activated receptor gamma (PPARγ) [52] and CCAAT-enhancer-binding proteins (C/EBPs) [53], or down-regulate the lipolytic ones. Modification in the sebum rates can be also consequent to the expression of perilipins such as PLIN2 and PLIN3, which coordinate lipogenic pathways, size and rate of lipid droplets formation [54,55]. The peripheral nervous system (PNS) likely plays a marginal or null role in SG regulation. Indeed, it is unclear if SGs, unlike the sweat glands, have an autonomic innervation [56]. Nevertheless, a role is played by substance P, owing to its pro-lipogenic and pro-inflammatory effects on SGs [57]. However, its immunoreactivity in cutaneous nerves near the SG has been seen only in glands associated with acne lesions [58]. Interesting studies showed that fibrillary aggregates of the protein alpha-synuclein (αSN) are not exclusive to the CNS, but they are present in other organs, including the skin. Specifically, αSN depositions have been found in dermal autonomic nerve fibers of PD, dementia with Lewy bodies (DLB) and pure autonomic failure (PAF) patients, and in unmyelinated somatosensory fibers of the sub-epidermal plexus in multiple-system atrophy (MSA) patients [59]. Cutaneous αSN is present in early PD stages, indicating that it may be present during the premotor stages [59]. Furthermore, αSN deposits are detected in cells from over 60% of pilosebaceous units of PD patients [60]. Thus, quantitative and qualitative lipids alterations seen in PD patients may be the consequence of αSN deposits in SG, which could directly interfere and alter sebum secretion and production. Moreover, SER and altered lipids sebum composition may be preclinical signs of PD [61]. Alterations could also be due to αSN-related autonomic dysfunction or, more likely, cutaneous αSN may modify the neuromediators release from the cutaneous autonomic nerve fibers, probably increasing substance P (similar to the alterations seen in acne, as mentioned above) and influencing SG secretion indirectly.

Ultimately, PD could be associated with a cutaneous microbiome alteration likely due to facial hypo-amimia, an early PD sign, and poor patient’s hygiene. This alteration may determine, by paracrine control, a change in lipids secretion and lipogenesis pathways of sebum. Indeed, some of the lipid targets found in PD sebum play an essential role in cutaneous homeostasis: myristic acid (C14:0) is important for antimicrobial response against several bacteria and fungi [62]; oleic acid (C18:1n-9) has bactericidal activity [63]; lignoceric acid (C24:0) is important for the maintenance of the skin barrier [64]. The altered cutaneous microbiome could also be responsible for FFAs incretion. Indeed, one of the most studied colonizing skin members is C. acnes which, through its lipases and peroxidases, is able to metabolize TGs in FFAs [46,65]. Sebum lipidomics studies showed that also the yeast Malassezia furfur, through its lipases and phosphatases, can cause the elevation of FFAs, which could irritate skin and promote development of SD [46]. Recently, it was found that the sebum of the skin of PD subjects is responsible for peculiar skin odor in association with a distinct volatiles signature [21,22]. This finding, together with our results, suggests that sebum may offer valuable information regarding the PD epiphenomena or pathogenetic mechanisms. Sebum lipidomics has shown some potential in identifying skin biomarkers also in AD. The deregulated association between TGs with FFAs is worthy of further investigations. Even though the relationship between serum TGs levels and cerebral amyloidosis or AD is controversial [66], studies on compositional changes of TGs in sebum and plasma are needed for a better characterization of AD stages. Furthermore, investigations during the preclinical phases coincident with subtle neuropathological changes are needed in both diseases to verify if SER and qualitative sebum alterations may be premonitory signs of NDDs.

This is a pilot study to be followed with investigations on sebum lipids changes on a larger cohort to understand if there is a link with the type and severity of AD and PD neurological manifestations or if SER and qualitative sebum alterations may be premonitory signs of NDDs. Thereby, the data are encouraging toward the use of sebum lipidomics as a non-invasive method for the research of biomarkers for NDD screening, diagnosis, and management.

## 4. Materials and Methods

### 4.1. Study Design and Participants

The present one is an observational case-control study conducted in three groups: 20 PD patients (age 71.5 ± 7.16 years), 20 AD patients (age 79.1 ± 6.03 years), and 24 age/sex-matched healthy subjects (HS) (age 72.2 ± 10.37 years) whose characteristics are described in Table 1.

The participants were consecutively enrolled among individuals accessing the Centre for Research and Training in Medicine of Aging (CeRMA) of the University of Molise (Campobasso, Italy). Eligibility criteria for AD patients were: diagnosis of “probable AD with documented decline”, according to criteria of the NIA-AA (National Institute on Aging and Alzheimer’s Association) [67], Mini-Mental State Examination (MMSE) < 24, and Clinical Dementia Rating (CDR) scale > 1. Eligibility criteria for PD patients were: diagnosis of “clinically established PD”, according to the MDS (Movement Disorder Society) criteria [68], dopaminergic therapy started at least 2 months prior to enrollment MMSE > 24; and CDR scale < 1. Exclusion criteria for AD, PD, and HS were: concomitant dermatologic pathologies such as acne, atopic dermatitis, psoriasis, other non-specific dermatitis, immunosuppression and HIV seropositivity. To rule out other potential causes, all AD and PD patients underwent blood tests (including complete blood count, erythrocyte sedimentation rate, urea and electrolytes, thyroid function, vitamin B12, and folate) and brain imaging.

### 4.2. Sebum Collection

According to a standardized procedure, the subjects who agreed to join the study underwent sampling of sebum with adhesive patches (Sebutape™, CuDerm Corporation, Dallas, TX, USA) applied onto the central region of the forehead. After sebum accumulation for 30 min, sebum net weight (µg) was calculated as the weight difference between patches before and after application. Sebum excretion rates (SER) were derived from sebum weight (µg), collection time (30 min) and sampling area (11.2 cm^2^) and were expressed as µg/cm^2^/minute. The patches were stored folded in a cryotube at −80 °C until extraction.

### 4.3. Materials, Chemicals, and Reagents

Authentic lipid standards were purchased from LARODAN (Malmo, Sweden), Cayman Chemical (Ann Arbor, MI, USA), Toronto Research Chemicals (TRC, North York, ON Canada), and C/D/N Isotopes (Pointe-Claire, QC, Canada) as detailed in Appendix A. The chemicals to prepare the solution of BSTFA in pyridine containing 1% trimethylchlorosilane (TCMS) were from Merck (Darmstadt, Germany).

### 4.4. Sample Preparation and Sebum Analysis

Sebum lipids were extracted from the tapes as previously reported [48,65]. Briefly, on the day of extraction, the adhesive patches were transferred to the glass tubes pre-loaded with 200 µL of the internal standard (iSTD) mixture (d6cholesterol 100 µM, d6squalene 100 µM, and d17C16:0 50 µM in isopropyl alcohol). The patches were extracted twice with 6 mL ethanol containing 0.0025% of butylhydroxytoluene (BHT) to prevent oxidation, and further processed by liquid–liquid extraction with ethyl acetate, as described [65]. The upper organic phases were pooled into pre-labeled clean glass tubes and dried under nitrogen. The dry extract was dissolved in 500 µL of the acetone/methanol/isopropanol (40/40/20 *v*/*v*/*v*) mixture to be analyzed by GCMS and TLC.

### 4.5. Sebum Lipid Profiling

Sebum lipidomics was performed by GCMS following the generation of trimethylsilyl (TMS) derivatives, as previously reported [48]. GCMS measured individual lipid species: 14 branched fatty acids (BCFAs), including 8 with even (eBCFA) and 6 with odd (oBCFA) number of carbons, 10 monounsaturated fatty acids (MUFAs), 2 polyunsaturated fatty acids (PUFAs), 15 straight chain saturated fatty acids (SCFA), including 8 and 7 with even (eSCFA) and odd (oSCFA) number of carbons, respectively. Moreover, 7 fatty alcohols (FOHs), squalene, cholesterol, and vitamin E were determined. For the quantitative analyses of individual sebum components determined by GCMS, we constructed calibration curves of the authentic standards listed in Appendix A. Amounts of putative known lipids were referred to the structurally closest compound available as an authentic standard as previously described [48]. In addition, TLC was used to quantify total amounts of triglycerides (TGs) and wax esters (WEs), which complemented the GCMS quantitative data. Amounts of TGs and WEs were determined against calibration curves of authentic TGs and WEs, loaded on the same plate together with the sebum samples as reported [48].

### 4.6. Data Analysis and Chemometric Modelling

Data were analyzed using the statistical and data analysis solutions XLSTAT 2020. 1. 2 (Addinsoft, New York, NY, USA), and MatLab (version 8.6.0 release R2015b; The Mathworks, Natick, MA, USA). Continuous variables were represented as average values with confidence intervals or mean ± standard deviation (SD). Prior to analysis, data were normalized through log-transformation, and standardized; two-tailed Student’s *t*-test and ANOVA were then used for comparison between two or more groups, respectively. Spearman’s coefficient (R) was used to measure the correlation between two quantitative variables. Differences and correlations were considered statistically significant with *p* ≤ 0.05 and *p* ≤ 0.017, respectively. The latter one was decided employing the Bonferroni correction, a multiple-comparison correction used when several dependent or independent statistical tests are being performed simultaneously. In order to avoid a lot of spurious positives, the probability alpha value is kept low to account for the number of comparisons being performed. To compare three groups with a desired probability alpha value ≤ 0.05 the Bonferroni correction would test each individual hypothesis at alpha = 0.05/3 = 0.017.

To approach the multivariate data output from GCMS and TLC together, we used ANOVA-simultaneous component analysis (ASCA). Combining analytical outputs with ASCA was applied to the search of sebum bio-signatures associated with HS, AD and PD. The main advantage of ASCA being the possibility to combine multivariate ANOVA decomposition of the experimental data matrix with non-parametric testing and interpretation of the multivariate effects using principal component analysis (PCA) [69,70,71,72], this test was applied to the data matrices on sebum. In detail, by indicating as X the matrix collecting the results of the designed experiments, the first step of the procedure was to partition the variability in it, according to the ANOVA scheme, i.e., as a sum of additive terms, each accounting for the effect of a particular design term (factor or interaction). In the present study, the data matrix is decomposed as follows:X = X_m_ + X_NDD_ + X_G_ + X_NDD&G_ + X_res_(1)
where NDD represented the AD, PD and HS conditions and G was the gender. All these matrices have the same dimension but are constructed in different ways. In X_m_, each row contains the mean experimental profile calculated on all the samples; indeed, this “grand mean” matrix is introduced to express the variation induced by the factor(s) as differences with respect to the mean profile. X_NDD_ and X_G_ are effect matrices with the sample estimates of the level means for the NDD and gender factors, respectively; X_NDD&G_ is the effect matrix containing the estimates of the interaction effect between NDD and gender, represented by the means of each NDD and gender combination after subtracting the means of both main effects; X_res_ is the residual matrix, which contains the variability not explained by the linear ANOVA model. Significance of an effect can be estimated by the sum of squares (SSQ) of the elements of the effect matrices (X_NDD_ and X_G_) and its statistical significance may be evaluated by comparing the experimental SSQ with its distribution under the null hypothesis, which can be obtained by permutation tests. If the design term is found to have a significant effect, then interpretation can be carried out by calculating a PCA model of its effect matrix, which allows evaluating the changes in the multivariate experimental profile induced by the different levels of the controlled factor(s).

## Figures and Tables

**Figure 1 metabolites-11-00819-f001:**
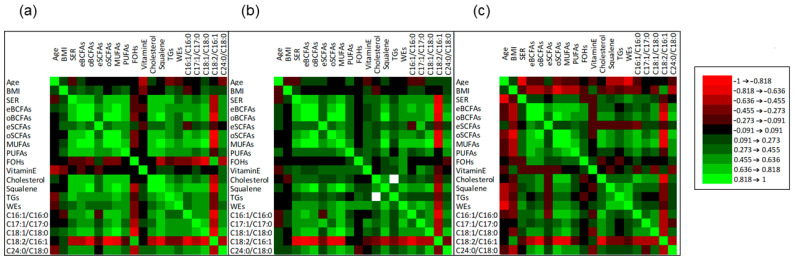
Matrices of Spearman’s correlation among age, BMI, SER, absolute amounts (µg) of grouped sebum lipids, and indexes of FA metabolism, i.e., FADS2 (C16:1/C16:0 and C17:1/C17:0 ratios), SCD1 (C18:1/C18:0 ratio), linoleate/sapienate ratio (C18:2/C16:1), and elongation (lignocerate/stearate ratio, C24:0/C18:0) in HS (**a**), AD (**b**), and PD (**c**) groups. FFAs were grouped in eBCFAs, oBCFAs, eSCFAs, oSCFAs, MUFAs, and PUFAs (see Appendix A for the labeling). FOHs is the sum of individual FOHs. Degree of correlation and signs are depicted according to the color grade in the legend. Correlation values and significance are reported in the Appendix A. Correlation significance cut-off was set at 0.017 (Bonferroni correction).

**Figure 2 metabolites-11-00819-f002:**
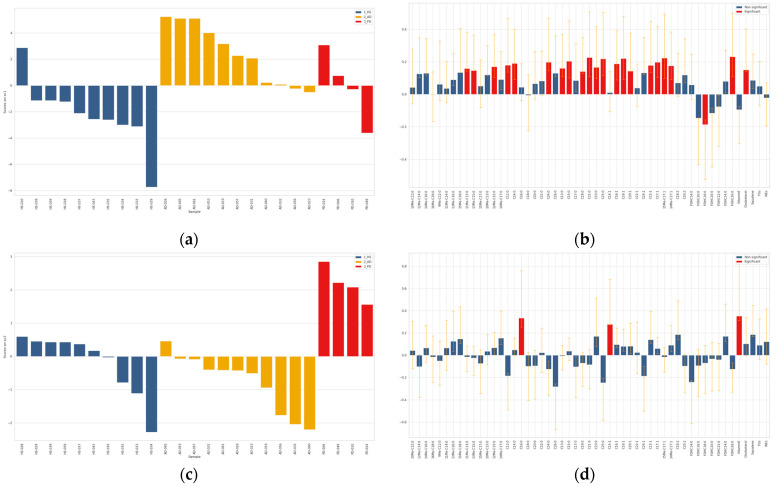
ASCA analysis on the effect matrix (absolute amounts, µg) for NDD conditions investigated in sebum from female (F) subjects. Panels (**a**,**c**) report the SC1 and SC2 scores, respectively, after projection of the residuals onto the space spanned by the significant principal component (PC) 1 and PC2; legend: blue = HS; yellow = AD; red = PD. Panels (**b**,**d**) display the variable loadings on SC1 and SC2, respectively, together with their confidence interval (red and blue bars indicate significantly and not significantly contributing descriptors, respectively).

**Figure 3 metabolites-11-00819-f003:**
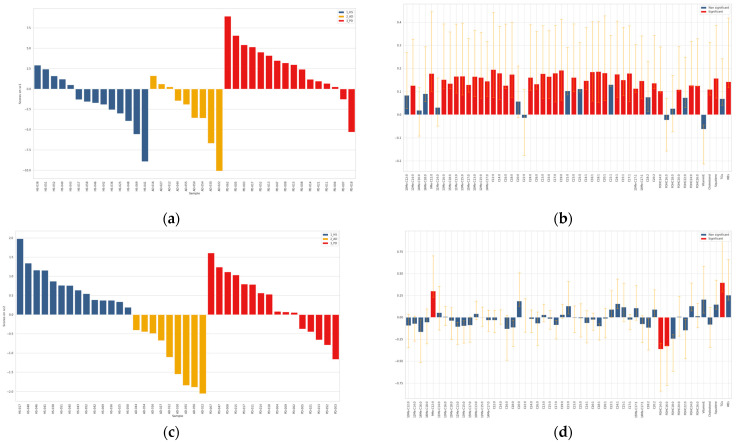
ASCA analysis on the effect matrix (absolute amounts, µg) for NDD conditions investigated in sebum from male (M) subjects. Panels (**a**,**c**) report the SC1 and SC2 scores, respectively, after projection of the residuals onto the space spanned by the significant principal component (PC) 1 and PC2; legend: blue = HS; yellow = AD; red = PD. Panels (**b**,**d**) display the variable loadings on SC1 and SC2, respectively, together with their confidence interval (red and blue bars indicate significantly and not significantly contributing descriptors, respectively).

**Table 1 metabolites-11-00819-t001:** Demographics, anthropometrics, lifestyle, education, motor function, and sebum excretion rates, in the three categories.

CATEGORIES	COUNTS	AGE (YEARS)	BMI (KG/M2)	SMOKING	ALCOHOL INTAKE	EDUCATION (YEARS)	MMSE	SER
**HS**	24	72.2 ± 10.4	26.6 ± 1.64	14 (58%)	12 (50%)	12.5 ± 3.79	29.8 ± 1.09	4.91 ± 2.76
**HS|F**	10	74.1 ± 8.27	26.2 ± 1.23	6 (60%)	3 (30%)	10.7 ± 3.8	30.3 ± 1.35	3.33 ± 1.44
**HS|M**	14	70.8 ± 11.8	26.9 ± 1.87	8 (57%)	9 (64%)	13.8 ± 3.33	29.5 ± 0.74	6.04 ± 2.96 ^#^
**AD**	20	79.1 ± 6.03	26.1 ± 3.56	6 (30%)	10 (50%)	8.35 ± 3.33 **	21.4 ± 3.20 ***	5.37 ± 2.68
**AD|F**	11	78.5 ± 6.17	25.1 ± 4.29	2 (18%)	5 (45%)	7.09 ± 2.95	21.5 ± 2.95 °°°	4.79 ± 2.67
**AD|M**	9	79.8 ± 6.14	27.3 ± 2.03	4 (44%)	5 (56%)	9.89 ± 3.26 ^#^	21.4 ± 3.67 °°°	6.09 ± 2.65
**PD**	20	71.5 ± 7.16	26.1 ± 2.50	8 (40%)	10 (50%)	11.3 ± 4.52	26.3 ± 1.99 **	8.05 ± 4.09 *
**PD|F**	4	72.0 ± 5.03	28.5 ± 2.02	1 (25%)	2 (50%)	10.5 ± 3.69	27.3 ± 1.21	5.51 ± 2.86
**PD|M**	16	71.3 ± 7.73	25.5 ± 2.28 ^#^	7 (44%)	8 (50%)	11.5 ± 4.79	26.1 ± 2.10 °	8.69 ± 4.17

BMI, body mass index; MMSE, Mini-Mental State Examination; SER, sebum excretion rate. Kruskal–Wallis test: * Significant differences among HS, AD, and PD groups independent on the sex (Bonferroni corrected significance level: 0.0167, * *p* ≤ 0.0167. ** *p* ≤ 0.00167. *** *p* ≤ 0.000167); ° Significant differences among HS, AD, and PD separately for F and M subjects (Bonferroni corrected significance level: 0.0033, ° *p* ≤ 0.0033. °°° *p* ≤ 0.000033); ^#^ Significant differences between F and M in the same group (^#^
*p* ≤ 0.05).

**Table 2 metabolites-11-00819-t002:** The table reports the amounts of sebum lipids (µg) quantified in sebum sampled in 30 min from two Sebutape patches. Results are reported as average ± standard deviation (SD). Mann–Whitney test was used for the comparison between HS and each AD and PD groups. Fold changes (FC) of averaged values in the AD or PD group vs. controls were in bold when *p*-values were <0.05.

	Average ± SD (HS)	Average ± SD (AD)	Average ± SD (PD)	FC AD vs. HS	*p*-Value	FC PD vs. HS	*p*-Value
10Me-C12:0	0.143 ± 0.159	0.164 ± 0.216	0.296 ± 0.292	1.152	>0.05	**2.07**	**0.036**
12Me-C14:0	1.38 ± 1.91	1.63 ± 2.06	4.98 ± 7.41	1.179	>0.05	**3.61**	**0.014**
14Me-C16:0	1.49 ± 1.93	2.31 ± 2.98	1.96 ± 1.36	1.538	>0.05	1.32	0.057
16Me-C18:0	0.851 ± 1.33	1.17 ± 1.81	1.96 ± 2.69	1.376	>0.05	2.30	>0.05
9Me-C12:0	0.111 ± 0.136	0.0488 ± 0.0442	0.393 ± 0.774	0.443	>0.05	3.54	>0.05
11Me-C14:0	0.357 ± 0.923	0.389 ± 0.773	0.387 ± 0.647	1.091	>0.05	1.08	>0.05
13Me-C16:0	1.59 ± 2.08	1.47 ± 1.16	5.01 ± 6.59	0.920	>0.05	**3.15**	**0.006**
15Me-C18:0	2.40 ± 2.63	2.54 ± 1.97	5.78 ± 4.77	1.057	>0.05	**2.41**	**0.008**
11Me-C13:0	0.365 ± 0.336	0.412 ± 0.297	0.969 ± 1.37	1.129	>0.05	**2.65**	**0.006**
13Me-C15:0	3.452 ± 3.37	4.21 ± 2.97	9.393 ± 15.2	1.220	>0.05	**2.72**	**0.004**
15Me-C17:0	0.996 ± 0.847	0.944 ± 0.524	1.92 ± 3.02	0.948	>0.05	1.93	>0.05
10Me-C13:0	1.17 ± 1.21	1.01 ± 0.864	2.45 ± 1.61	0.859	>0.05	**2.09**	**0.010**
12Me-C15:0	9.34 ± 10.1	8.84 ± 7.24	19.8 ± 12.4	0.947	>0.05	**2.12**	**0.004**
14Me-C17:0	3.83 ± 3.08	3.56 ± 2.31	6.52 ± 3.42	0.929	>0.05	**1.70**	**0.010**
C12:0	3.32 ± 2.34	4.17 ± 2.51	6.09 ± 3.36	1.256	>0.05	**1.83**	**0.003**
C14:0	36.1 ± 39.6	38.1 ± 31.4	91.3 ± 73.2	1.057	>0.05	**2.53**	**0.001**
C16:0	136 ± 120	159 ± 123	291 ± 212	1.171	>0.05	**2.14**	**0.002**
C18:0	41.5 ± 27.7	41.3 ± 17.2	49.5 ± 20.1	0.996	>0.05	1.19	>0.05
C20:0	2.63 ± 2.21	2.21 ± 1.45	3.63 ± 3.91	0.842	>0.05	1.38	>0.05
C22:0	2.12 ± 1.61	1.94 ± 0.711	2.72 ± 2.65	0.915	>0.05	1.28	>0.05
C24:0	3.16 ± 2.92	3.49 ± 1.92	8.52 ± 8.44	1.102	>0.05	**2.70**	**0.015**
C26:0	0.556 ± 0.686	0.631 ± 0.621	1.68 ± 1.98	1.133	>0.05	3.02	>0.05
C13:0	1.17 ± 1.186	1.23 ± 1.21	2.47 ± 1.61	1.054	>0.05	**2.11**	**0.002**
C15:0	26.7 ± 28.9	29.7 ± 24.8	65.3 ± 51.6	1.112	>0.05	**2.45**	**0.003**
C17:0	8.07 ± 7.16	8.72 ± 6.95	14.3 ± 9.45	1.081	>0.05	**1.77**	**0.007**
C19:0	0.989 ± 0.824	1.01 ± 0.686	2.07 ± 1.45	1.016	>0.05	**2.09**	**0.004**
C21:0	0.483 ± 0.677	0.329 ± 0.137	0.722 ± 0.824	0.682	>0.05	**1.49**	**0.048**
C23:0	0.516 ± 0.443	0.496 ± 0.233	1.14 ± 0.862	0.961	>0.05	**2.21**	**0.002**
C25:0	0.437 ± 0.424	0.576 ± 0.471	1.202 ± 2.176	1.316	>0.05	2.75	0.054
C14:1	0.934 ± 1.164	0.901 ± 1.012	2.75 ± 2.74	0.964	>0.05	**2.94**	**0.003**
C16:1	21.6 ± 25.8	24.1 ± 19.5	67.01 ± 67.2	1.117	>0.05	**3.10**	**0.001**
C18:1	38.7 ± 35.9	45.9 ± 27.3	92.2 ± 60.5	1.189	>0.05	**2.38**	**0.000**
C20:1	1.33 ± 1.22	1.39 ± 1.19	3.07 ± 2.14	1.052	>0.05	**2.31**	**0.002**
C22:1	0.144 ± 0.150	0.101 ± 0.0702	0.326 ± 0.351	0.700	>0.05	2.26	0.054
C24:1	0.283 ± 0.238	0.241 ± 0.210	0.641 ± 0.523	0.850	>0.05	**2.26**	**0.018**
C15:1	1.06 ± 1.31	1.09 ± 1.149	2.82 ± 2.95	1.032	>0.05	**2.66**	**0.004**
C17:1	4.19 ± 4.22	4.75 ± 3.96	10.4 ± 7.56	1.135	>0.05	**2.48**	**0.001**
15Me-C17:1	0.881 ±1.76	0.462 ± 0.277	1.26 ± 1.62	0.524	>0.05	**1.43**	**0.019**
14Me-C17:1	1.05 ± 1.19	1.02 ± 0.767	2.19 ± 1.37	0.968	>0.05	**2.09**	**0.003**
C18:2	3.19 ± 1.53	3.53 ± 1.59	4.33 ± 1.65	1.109	>0.05	**1.36**	**0.030**
C20:2	0.471 ± 0.501	0.451 ± 0.452	1.11 ± 1.04	0.956	>0.05	**2.36**	**0.048**
FOHC14:0	1.37 ± 0.908	3.91 ± 5.85	1.83 ± 1.31	2.859	>0.05	1.34	>0.05
FOHC16:0	10.4 ± 15.4	15.1 ± 23.9	7.17 ± 19.2	1.439	>0.05	0.69	>0.05
FOHC18:0	23.7 ± 26.6	23.8 ± 30.1	16.2 ± 26.3	1.007	>0.05	0.68	>0.05
FOHC20:0	7.29 ± 12.1	3.63 ± 3.19	4.21 ± 1.59	0.499	>0.05	0.58	>0.05
FOHC22:0	7.74 ± 13.3	5.95 ± 13.4	3.76 ± 1.71	0.769	>0.05	0.49	>0.05
FOHC24:0	1.82 ± 0.992	1.57 ± 0.568	2.59 ± 1.19	0.861	>0.05	**1.42**	**0.026**
FOHC26:0	0.919 ± 0.998	1.05 ± 0.872	2.67 ± 3.149	1.144	>0.05	**2.91**	**0.030**
Vitamin E	0.0271 ± 0.0776	0.0031 ± 0.0049	0.0062 ± 0.014	0.115	>0.05	0.23	>0.05
Cholesterol	17.1 ± 11.03	17.7 ± 7.85	22.1 ± 5.36	1.033	>0.05	**1.29**	**0.008**
Squalene	276.4 ± 252.9	202.1 ± 165.9	481.9 ± 241.8	0.731	>0.05	**1.74**	**0.010**
TGs	539.8 ± 315.7	415.9 ± 286.9	620.4 ± 299.6	0.771	>0.05	0.11	>0.05
WEs	361.2 ± 182.1	278.5 ± 132.9	496.7 ± 217.6	0.771	>0.05	**1.38**	**0.039**

## Data Availability

The datasets generated and analyzed during the current study are available from the corresponding author and the last author. The data are not publicly available due to their inclusion into the institutional repository.

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
