# Peer review of "Application of Sebum Lipidomics to Biomarkers Discovery in Neurodegenerative Diseases"

_metabolites, 2021, doi:10.3390/metabo11120819_

Round 1

Reviewer 1 Report

The authors presented a thorough study of detecting sebum biomarkers to distinguish between AD and PD from healthy individuals. This can have a lot of potential in detecting early onset of AD and PD. However I have a question for them: 

Why is PD associated with hyperseborrhea? Does gender also plays a role in this association?

Does alcohol intake has any effect on the lipid profile in sebum?

There is a typo in line 7 on Page 3. 

Author Response

Reviewer 1

Comments and Suggestions for Authors

The authors presented a thorough study of detecting sebum biomarkers to distinguish between AD and PD from healthy individuals. This can have a lot of potential in detecting early onset of AD and PD. However, I have a question for them:

Why is PD associated with hyperseborrhea? Does gender also plays a role in this association?

The reviewer poses two important questions that are worth of further investigations.

The mechanisms underlying hyperseborrhea in PD are still elusive. PD is associated with seborrheic dermatitis (SD), which, in turn, is intertwined with hyperseborrhea. Whether or not hyperseborrea bridges PD to SD awaits clarification. Both PD and SD are more prevalent in the male gender, suggesting to search underlying factors among sex-related mechanisms.

In the discussion we addressed the question with: Elevation of SER in PD can be due to delayed decline of the SG activity or to hyperactivation of the SG due to PD pathomechanisms sustaining sebogenesis, including derangement of regulatory factors in the neuroendocrine system in the CNS, and dysbiosis. Clarifying the temporal changes of excessive sebum production might have important implications in the detection of PD ahead of deterioration of the motor functions.

On the other hand, the response of the sebaceous gland to perturbed circulating lipids has been little investigated.

Does alcohol intake has any effect on the lipid profile in sebum?

Skin surface lipids, which include sebum, are suitable reporters of the alcohol abuse by means of fatty acid ethyl esters detection. Nevertheless, abundance of ethanol-fatty acid adducts has proven not to discriminate between non-drinkers and social drinkers in previous studies. In our study the percentage of social drinkers was consistently represented in the three groups examined. Thus, no correlation between alcohol intake and sebum parameters was attempted.

There is a typo in line 7 on Page 3.

The manuscript has been thoroughly revised. The typo should be fixed in the revision.

Reviewer 2 Report

  • I read the paper “Application of sebum lipidomics to biomarkers discovery in neurodegenerative diseases”, which is aimed to explore a possible correlation between lipid components of sebum and neurodegeneration associated with both AD and PD.

    This study evaluates sebum lipidomic profiles in two presumed NDD conditions.

    The parameters obtained with gas chromatography-mass spectrometry (GCMS) are mainly free fatty acids (FFA) integrated with squalene, cholesterol, vitamin E, triglycerides and wax esters, the latter determined by the thin layer chromatography (TLC). The results of the study may contribute to the knowledge of modified sebogenesis in PD.

    This study presents points of interest. However, it has some criticisms, which should be corrected.

    1. The most apparent function of sebaceous glands is to excrete sebum. Under these experimental conditions what is the contribution to be attributed to PD rather than AD (preclinical status).
    2. Although many efforts have been made to evaluate possible biomarker for AD, a definitive diagnosis is currently not applicable. The authors only on one occasion do they speak of “preclinical” AD, a discussion about this issue should be developed.

    3, Recently, a causative relationship between neurodegenerative and vascular mechanisms more frequent in patients with dementia has been highlighted. Triglyceride levels appear to be involved in cognitive function through putative mechanisms such as brain blood brain barrier dysfunction or amyloid metabolism imbalance, but not all research in the field has found this association. If further parameters were assessed the relationship between different forms of cognitive decline and serum triglyceride levels, independently of other cardiovascular risk factors, this would lead to hypothesize a role of triglycerides in cognitive decline, cerebral amyloidosis and vascular damage.

    In light with observation the possible association with other parameters should be evaluated and discussed.

Author Response

Reviewer 2

Comments and Suggestions for Authors

I read the paper “Application of sebum lipidomics to biomarkers discovery in neurodegenerative diseases”, which is aimed to explore a possible correlation between lipid components of sebum and neurodegeneration associated with both AD and PD.

This study evaluates sebum lipidomic profiles in two presumed NDD conditions.

The parameters obtained with gas chromatography-mass spectrometry (GCMS) are mainly free fatty acids (FFA) integrated with squalene, cholesterol, vitamin E, triglycerides and wax esters, the latter determined by the thin layer chromatography (TLC). The results of the study may contribute to the knowledge of modified sebogenesis in PD.

This study presents points of interest. However, it has some criticisms, which should be corrected.

  1. The most apparent function of sebaceous glands is to excrete sebum. Under these experimental conditions what is the contribution to be attributed to PD rather than AD (preclinical status).

We agree that the preclinical status of AD has become a major research focus since early intervention may offer the best chance of therapeutic success. As mentioned in the paper, this was a pilot study carried out in patients with overt disease. Therefore, we can’t address the question with the present data.

  1. Although many efforts have been made to evaluate possible biomarker for AD, a definitive diagnosis is currently not applicable. The authors only on one occasion do they speak of “preclinical” AD, a discussion about this issue should be developed.

As reported in the previous point, we agree that the preclinical status of AD has become an important research focus. In the last paragraph of the paper, we now added that: investigations on preclinical stages of both diseases, when an individual appears normal but the typical neuropathological changes are already developing, are needed to verify if SER and qualitative sebum alterations may be premonitory signs of NDDs.

  1. Recently, a causative relationship between neurodegenerative and vascular mechanisms more frequent in patients with dementia has been highlighted. Triglyceride levels appear to be involved in cognitive function through putative mechanisms such as brain blood brain barrier dysfunction or amyloid metabolism imbalance, but not all research in the field has found this association. If further parameters were assessed the relationship between different forms of cognitive decline and serum triglyceride levels, independently of other cardiovascular risk factors, this would lead to hypothesize a role of triglycerides in cognitive decline, cerebral amyloidosis and vascular damage. In light with observation the possible association with other parameters should be evaluated and discussed.

This is an important point raised by the reviewer. In the discussion section, we added that the relationship between serum TGs levels and cerebral amyloidosis or AD is controversial [Dimache et al, 2021]. Indeed, looking at the correlation data, triglycerides in AD lack of significant association with the levels of other sebaceous lipids, opposite to the HS group (more extensive association) and to PD (to a lesser extent). The levels of TGs tended to be lower in AD, however the statistical significance was not reached. Due to the subtle changes and the sample size, this is of course a starting point to extend investigation to a larger cohort of subjects.

Additions to the text:

Page 5: Due to the MUFAs-TGs and MUFA-WEs interdependency in sebaceous lipid pathways, we evaluated the MUFAs to TGs ratio and MUFAs to WEs ratios in the HS and NDD conditions. The MUFAs/TGs ratio was 1.8 and 3.2-fold higher in AD and PD, respectively, compared to HS; differences were close to significance (p=.017) and significant (p=.002) in AD and PD, respectively (data not shown). The MUFAs/WEs ratio was 1.5 and 2.0-fold higher in AD and PD, respectively, compared to HS; divergence from HS were significant in both AD and PD, being p=.008, and p=.002, respectively (data not shown).

Page 6: Amounts of TGs presented several positive and significant correlations with abundance of summed sebaceous-type FFAs in HS. Such correlations were absent in both AD and PD, suggesting that the TGs composition could be uncoupled with the abundance of FFAs in these NDDs. Interestingly, while the abundance of TGs correlated with that of squalene and WEs in PD subjects, consistently with HS, TGs lacked association with any other sebum-parameter in AD.

Page 11: TGs account for the highest w/w % in sebum. TGs are formed upon esterification of glycerol with FAs. Abundance and composition of TGs in sebum are associated to the variety of sebaceous-type FFAs. The association between sebaceous FFAs and TGs observed in unaffected controls was apparently deregulated in both NDDs. Together with the observa-tion of higher MUFAs/TGs and MUFAs/WEs ratios in both NDDs, these features were convergent in AD and PD sebum.

Page 12: Sebum lipidomics has shown some potential in identifying skin biomarkers also in AD. The deregulated association between TGs and FFAs is worth of further investigations. Even though the relationship between serum TGs levels and cerebral amyloidosis or AD is controversial [72], studies on compositional changes of TGs in sebum and plasma are needed for a better characterization of AD stages. Furthermore, investigations during the preclinical phases coincident with subtle neuropathological changes are needed in both diseases to verify if SER and qualitative sebum alterations may be premonitory signs of NDDs.